# Adaptive coping strategies among individuals living with long-term chikungunya disease: a qualitative study in Curaçao

Churnalisa Doran ,[1] Ashley J Duits,[2] Izzy Gerstenbluth,[3] Adriana Tami ,[1] Ajay Bailey[4]

[1]Department of Medical Microbiology and Infection Prevention, University Medical Center Groningen, Groningen, The Netherlands
[2]Department of Immunology, Curaçao Biomedical and Health Research Institute, Willemstad, Curaçao
[3]Department of Epidemiology, Curaçao Biomedical and Health Research Institute, Willemstad, Curaçao
[4]Department of Human Geography and Spatial Planning, University of Utrecht, Utrecht, The Netherlands

**Correspondence to**
Churnalisa Doran;
c.e.doran@umcg.nl

## ABSTRACT

**Objective** Long-term chikungunya disease is characterised by persistent rheumatic symptoms following chikungunya virus infection. As there is no specific treatment available, affected individuals need strategies to adapt. However, research on these is scarce. This study aimed to explore which adaptive coping strategies are employed to manage persistent rheumatic symptoms in daily life.

**Setting** The study was conducted in Curaçao.

**Design and participants** An explorative qualitative study was conducted between September and October 2020, among a purposive sample of adults, 19 women and 4 men affected by long-term chikungunya disease. In-depth interviews were semi-structured and transcribed verbatim. The data were analysed using inductive thematic analysis.

**Results** The disease duration for all participants ranged between 68 and 74 months (6 years). In narrating their experiences of coping with long-term chikungunya disease, four themes were identified: (1) learning to live with the disease; (2) resilience for dealing with pain; (3) maintaining positive self-image and attitude; and (4) coping through spirituality.

**Conclusion** To live with long-term chikungunya disease with dignity in spite of physical pain and discomfort, participants tried to retain a sense of control of oneself and one's lives, to not let the disease take over, focusing on the positive in their lives, and finding strength and remain hopeful. Interventions such as cognitive–behavioural therapy and mindfulness exercises may be effective in strengthening or regain affected individual's sense of competence and control by fostering adaptive coping skills and resilience. Subsequently, these interventions may improve health-related quality of life when rheumatic symptoms persist following chikungunya virus infection.

## STRENGTHS AND LIMITATIONS OF THIS STUDY

⇒ This is the first study to explore coping strategies among individuals affected by long-term chikungunya disease, 6 years after disease onset.

⇒ The explorative qualitative study design supported in the understanding of coping strategies more broadly and embedded in the cultural context.

⇒ All participants were Afro-Caribbean and mostly female, findings may, therefore, not be generalisable to a wider population.

⇒ While symptoms were not medically confirmed, they agreed with the current knowledge of long-term chikungunya disease sequelae.

⇒ Our study highlights the need to explore coping strategies among lower socioeconomic populations, who may face even more challenges in living and coping with long-term chikungunya disease.

## INTRODUCTION

Chikungunya is a re-emerging arboviral disease caused by the chikungunya virus (CHIKV) and represents a growing public health problem.[1] Over the past two decades, over 10 million cases have been reported,[2] with an estimated 1.3 billion people living in regions at risk of CHIKV transmission.[3]

Long-term chikungunya disease, characterised by persistent rheumatic symptoms of moderate to severe pain intensity including disabling polyarthralgia, joint swelling and stiffness, and myalgia of an unpredictable and recurrent nature, occurs in more than 30% of individuals following CHIKV infection.[1 4–7] In addition, long-term chikungunya disease is associated with emotional distress, seriously impacting daily lives and both physical and psychological health-related quality of life (QoL) of those affected.[8–11] In the absence of effective cures and treatment options,[12 13] improving coping strategies that can improve QoL becomes a critical objective in fostering the ability to live a valued life despite persistent symptoms. Coping can be defined as an individual's cognitive and behavioural efforts employed in response to stressors that determine how these stressors will affect their physical and emotional well-being.[14] Adaptive coping strategies are effective at benefiting or positively affect the lives of individuals who

use them, including their functionality and psychological well-being.[15]

Understanding of long-term chikungunya disease has been dominated by a singular focus of physical symptomology.[1 5 16 17] To our knowledge, to date, only one study explored coping strategies employed by individuals from acute disease onset to 1-year postinfection.[18] Thus, more research is needed to elicit information about the employed coping strategies individuals affected by long-term chikungunya disease perceive to be adaptive and useful,[19] which may help identify interventions directed at enhancing adaptive behaviours and well-being. The aim of this study was to explore the adaptive coping strategies employed by individuals affected by long-term chikungunya disease, 6 years after disease onset, in Curaçao.

## METHODS
### Study setting and design
Curaçao is an island located in the southern Caribbean Sea. Curaçao is culturally diverse with more than 60 different nationalities, with the largest ethnic group belonging to the Afro-Caribbean ethnicity and Roman Catholics comprising 70% of the population.[20 21] In June–July 2014, a CHIKV epidemic became evident, infecting approximately 30%–50% of the 150 000 inhabitants.[22]

This explorative qualitative study was part of a larger prospective longitudinal chikungunya cohort study of individuals with confirmed CHIKV infection.[22] Qualitative research designs are appropriate when the aim is to gain in-depth understanding of experiences and perspectives of individuals.[23] The study followed the Criteria for Reporting Qualitative Research (COREQ) checklist (online supplemental file 1).[24]

### Patient and public involvement
Our data collection tool was piloted to ensure that it is inclusive and comprehensive, through in-depth interviews with four individuals affected by long-term chikungunya disease recruited from the public.

### Participants and recruitment
Twenty-three participants were recruited from the cohort study.[22] Maximum variation purposive sampling using cohort records from a follow-up study conducted between July 2019 and March 2020 was employed to identify potential participants based on persistent rheumatic symptoms (arthralgia, swelling, stiffness, cramps and/or locking in one or more joints and/or myalgia), pain intensity score (0–10) of at least 1, gender and different age, to allow for a wide range of experiences and adaptive coping strategies employed.[25 26] In September 2020, 51 women and 10 men who self-declared being affected by persistent rheumatic symptoms following CHIKV infection were identified and considered potentially eligible. The eligible patients were contacted by CD (first author) and invited to participate by telephone on the basis they met the inclusion criteria of still experiencing persistent rheumatic symptoms, at

the time of contact, if not they were excluded. Out of the 29 individuals (19 women and 10 men) approached, 6 men declared being recovered from rheumatic symptoms and were excluded. Participant inclusion continued until data saturation was met during data collection. In total, 19 women and 4 men participated, received information about the study aim and signed a written informed consent. The disease duration ranged between 68 and 74 months (6 years).

### Data collection
Data collection was undertaken from September to October 2020. In-depth interviews were conducted in the native local language Papiamentu by CD, a female PhD candidate interested in infectious diseases with experience conducting interviews and who is fluent in both Papiamentu and English. Participants were made aware that CD was leading the cohort study, in which she had professional contact with three of the included participants. The interviews took place in a one-on-one meeting at the time and place that suited the participants. Interviews were guided by a piloted semistructured topic guide, developed by CD based on the literature review that identified important open-ended questions on adaptive coping strategies. The topic guide was discussed and evaluated by AB (senior author) a senior qualitative researcher (online supplemental file 2). The interviews lasted between 25 and 75 min and were audiorecorded. Participants were asked to discuss their experiences and coping strategies in response to persistent rheumatic symptoms. Data saturation was reached after 19 interviews, with the last 4 interviews being coded for confirmation.[27] Saturation was considered to have been met when probing gained no additional information or insights, signifying that further data collection is redundant and an adequate sample size has reached.[28] Demographic data related to educational status were extracted from cohort records. The interviews were transcribed verbatim by an independent transcriber, native in Papiamentu. Transcripts were cross-checked with original audiorecordings by CD.

### Data analyses
The profile of the participants including the purposive sampling criteria and demographic data was not considered to structure or analyse the narratives. The data were analysed using thematic analysis. Thematic analysis is data driven, and therefore, does not require a pre-existing theoretical framework, unlike any other methods for qualitative data analysis.[29] The narrative data were analysed in Papiamentu to preserve linguistic meanings. For familiarisation, individual transcripts were read and re-read several times by CD, followed by line-by-line open codification and continuous comparison of initial codes within and between transcripts. Two randomly selected transcripts which were particularly rich in data were translated to English by CD and independently checked and analysed by AB. Subsequently, themes were identified and analysed by repeated revision of the transcripts,

**Table 1** Sociodemographic and rheumatic characteristics of participants

| Characteristics | N (total=23) |
|---|---|
| Gender | |
| Female | 19 |
| Male | 4 |
| Age range (year) | |
| 32–49 | 9 |
| 50–65 | 12 |
| >65 | 2 |
| Relationship status | |
| Single/divorced/widowed | 7 |
| In relationship/married | 14 |
| Unknown | 2 |
| Educational status (highest degree obtained)* | |
| Low | 2 |
| Medium | 15 |
| High | 6 |
| Employment status | |
| Unemployed/homemaker | 5 |
| Paid work | 15 |
| Retired | 3 |
| Rheumatic symptoms† | |
| Arthralgia | |
| Upper extremities | 4 |
| Lower extremities | 4 |
| Both upper and lower extremities | 11 |
| Arthritis (joint swelling and/or stiffness) | |
| Upper extremities | 3 |
| Lower extremities | 6 |
| Both upper and lower extremities | 2 |
| Joint weakness | |
| Upper extremities | 4 |
| Lower extremities | 0 |
| Both upper and lower extremities | 4 |
| Joint cramps or locking | |
| Upper extremities | 3 |
| Lower extremities | 1 |
| Both upper and lower extremities | 2 |
| Myalgia | 1 |

*Categorised into low (primary education or less), medium (higher secondary education) and high (bachelor or masters).
†More then one answer was possible.

until themes saturation.[27] The analysis process followed an inductive and iterative approach, which is a process of coding the data without trying to fit it into a pre-existing theoretical frame.[30] At regular time intervals, CD and AB worked together to reach consensus on the codes and themes to reduce potential subjectivity and ensure credibility and trustworthiness.[31] The Atlas.ti programme was used to facilitate the iterative process. Relevant quotes translated to English by CD are presented to illustrate key themes. The quotes were grammatically altered to improve readability when necessary.

## RESULTS

Participants had all experienced persistent rheumatic symptoms on a daily or weekly basis. Table 1 summarises participants' characteristics. Analysis of the interviews revealed numerous coping strategies that were organised into four themes: learning to live with the disease, resilience for dealing with pain, maintaining a positive self-image and attitude, and coping through spirituality. Online supplemental table 1 presents the code list with codes and additional illustrative quotes related to the themes.

### Theme 1: learning to live with the disease

The majority of participants reported that they had to learn living with long-term chikungunya disease over the years, mainly by accepting the fact that the rheumatic symptoms could be a constant presence in their lives 'if it need to stay, it will stay' (Pt.9). Participants described that there was no other choice but accepting the disabilities and functional impairment as part of a new reality.

> It is already here [persistence of symptoms]. […]. You [talking in third person] need to learn to live with it [persistence of symptoms], accept. […]. If you can accept it, I think you can live faster with it, you can cope faster with it if you accept it. If not, your life will be miserable. (Pt.7)

> You [talking in third person] are not healthy [due to disabilities], because you need to lift up your leg to be able to walk [which is now difficult due to physical limitations], so you do not feel 100%–100% [fully healthy]…but yes, you have to live with it [disabilities], there is no other way. (Pt.14)

Remaining positive and knowing that chikungunya disease is non-life-threatening helped in the acceptance process.

> I am a person that is very positive, I am very positive. I will say…I know for sure that it is not a pain that will kill me. […]. I know why I have the pain in my leg, so I just accept it. (Pt.1)

> It is not that I got a viral disease that can take my life, it could have been worse. (Pt.11)

Participants described that the best way to live with the disease is to face the accompanying physical and emotional challenges to minimise the risk of psychological impairment.

> I say this has happened to me, I cannot break iron with my hands [trying to do something that is impossible] and it has already happened [persistency of

symptoms]. If I continue crying and whining about it then I think I will fall into stress, in a frustration, I will fall into a depression. (Pt.19)

Learning to live with long-term chikungunya disease also required balancing physical activities without over-straining. For these participants staying in control in the face of pain meant accepting more realistic standards and focus on the things they could do. They accepted their new functional levels as adequate and as a result, decreased the emotional fallout from not being able to perform at previous levels.

I do not have control over the symptoms. What I can control is living, taking consideration with myself with what I can do and what not…And that is what I spend my energy on, so I will not look at what I could do back in the days [pre-infection] and cannot do any more. […]. Because of that I have noticed that it does not hurt me [emotionally]. (Pt.23)

It does not need to become a trauma for you [talking in third person] that you cannot walk [conditional walk]. […]. When I cannot walk [due to flare-ups], I will often stand at my kitchen sink. […]. I will pretend that I am walking [by doing stepping motions]. (Pt.7)

### Theme 2: resilience for dealing with pain

Participants tried to live their lives in dignity in spite of persistent rheumatic symptoms. The majority developed resilience to the pain sensations, which gave a sense of empowerment and control. Participants described not giving in to the pain, 'I will not let the pain hamper me in anything that I want to do' (Pt.16). These participants described being prepared to negotiate discomfort and pain rather than adapt.

I will not let the pain take control of me. If I keep standing still, I will do nothing anymore. If I have to do it [something], I will do it normally, with or without pain. It [pain] hits me, but I will continue normally…I need to fight it [participant is laughing]. (Pt.13)

The next day after going out [dancing], I will have more joint pain or stiffness…but I do not care, that moment [going out] is more enjoyable for me. I will accept the pain, I feel good, because I did what I wanted to do [dancing]. (Pt.17)

Participants described the interconnectedness of the mind and body in symptoms aggravation and control 'if you keep on thinking about it, it will not go [you will keep on feeling it]' (Pt.5). Hence, conscious attempts were made to ignore pain sensations, driven by the idea that being aware of the pain would make it worse.

I switch my mind, I will not continuously think of the pain, because I belief that you will get more pain if you keep on thinking about the pain. It is the same when someone has the flu, they are sick and will think

oh my throat, oh my throat…oh my head, oh my this. No! I will think that I am okay. (Pt.14)

### Theme 3: maintaining positive self-image and attitude

The significance of remaining positive was a common thread. Participants emphasised the importance of putting things in perspective to maintain a positive self-image. Some described that practising cognitive techniques such as positive self-statements, they were able to cope with persistent rheumatic symptoms by shutting out negative or destructive self-thoughts. These statements included things that represented meaning and joy in their lives.

I have two daughters, that is what cheers me up. […]. It gives me meaning in life without whining about the pain. (Pt.6)

The process is also that you have to think to love yourself…Yes, when I think bad, I will think the good, I will say to myself…[name] you are beautiful, you have three children, you have to get well to see your grandchild grow up…And I think beautiful things and a happiness will come. (Pt.15)

Other cognitive approaches included comparing socioeconomic status and health with others that were perceived as worse off by downward comparison, thus minimising the severity of their own situation. This brought something positive and redeeming out of the persistency of symptoms.

Like I already said, there are people that are in a worse situation and I have learned to be thankful for what I have…My strategy is that I can walk, I can move, I have a life, I have a job, I have a house, I have a car, so there are people that are worse…Like there are people that their residuals [persistent symptoms] are worse compared to what I have. (Pt.3)

I have a friend that has multiple scleroses. She receives welfare, she lives in a dilapidated house, she has the disease [multiple sclerosis], she has a walking stick, she cannot do this, she cannot do that….but she is happier than you and me. Without her knowing, she taught me to see that not because you have a disease your life has stopped, persevere and keep on being positive. It has changed me allot. (Pt.19)

### Theme 4: coping through spirituality

To maintain being strong and hopeful in the uncertainty of ever returning to pre-chikungunya health status, participants turned to spirituality 'I have faith [in God] that the symptoms will go away' (Pt.4). Relating to their childhood upbringing, the majority believed that trusting in God would help through difficult times across their life course. This helped to find the strength and courage to endure long-term chikungunya disease.

God is the one that gives us strength to carry everything [every difficulty]. […]. I do not believe that

there is a pain that we feel that is more severe than the one that Jesus had to carry. (Pt.1)

I am a person that loves to pray. I will ask God to help me and release the pain from my body. That gives me the courage to continue. (Pt.7)

Participants also felt empowered from their prayers believing that through these prayers and practising faith, they had a sense of control over the disease.

I pray to God everyday, sometimes I ask God to feel his presence. I will ask that before getting out of bed. I will feel like he [God] strokes my knee with his hands, and I will get out of bed with more speed [less impaired]. (Pt.6)

I grew up with always belief, belief, belief, use your faith and hold on to your faith… you may have something, but it will not stay forever. It can take a while, but it will go. That is the hope and faith that I have. (Pt.17)

Some participants also believed that God had the power to guide effective treatments by acting through physicians. These participants expressed hope that the guidance would provide a cure for their symptoms. This introduced a hopefulness in their illness trajectory.

Only God can help us and he [God] gives the doctors the knowledge to help us…and the doctor has helped me. The syringe that he [doctor] gave me in my knee [intra muscular] has given me the strength to walk normal. (Pt.15)

## DISCUSSION

This explorative qualitative study draws on a sample of participants affected by long-term chikungunya disease to provide in-depth insights on employed adaptive coping strategies. Participants perceived persistent rheumatic symptoms beyond their control. Hence, they tried to preserve a life within the limits of symptom by addressing areas of their lives and oneself that could be controlled or influenced. This included learning to live with the disease, resilience for dealing with pain, maintaining a positive self-image and attitude, and coping through spirituality.

In this study, participant's narratives reflected an acceptance of persistent rheumatic symptoms as a facilitator to adjust and learning to live with the chronicity of the disease. The disease was embraced as a new reality and part of their life to get back in charge, consistent with previous published research.[32 33] In particular, Sturgeon and Zautra[33] concluded that pain acceptance can be a positive management technique which helps individuals to focus on factors that can be controlled, rather than the symptoms which are uncontrollable, hence enhancing a sense of empowerment.[33] Large body of research has demonstrated the positive effect of pain acceptance in health outcomes in chronic pain disorders such as

rheumatoid arthritis,[34] fibromyalgia,[35] cervical pain and low back pain.[36] Individuals with greater pain acceptance showed less pain related anxiety, performed activities of daily living with better efficacy and committed to achieve valued goals, showed improved physical and psychological QoL.[18 34 37–39] Participants restructured their expectations realistically and accepted more feasible physical goals. Pursuing achievable goals have been consistently identified as an important component of adaptation increasing perceived control and psychological well-being in chronic disease literature.[40–42]

Our findings complement other studies and provide further evidence that individual's adopting resilience to pain possess a great belief that they can effectively control their lives.[33 43] Similar to a rheumatoid arthritis study, bearing the pain and discomfort triggered by a particular activity was preferred to long-term chikungunya disease imposing constraints and adaptations on participant's lives.[44] In qualitive research on chronic pain conditions choosing to ignore the impact of persistent symptoms in activity engagement provided a means for individuals to move pain to the background of their consciousness and forget about their suffering and maintain a degree of normalcy.[45 46] These narratives reflect the components of acceptance suggested by McCracken[36]: acceptance of disease chronicity, pain-willingness and activity engagement.[36]

Some chronic illness literature described that the body is not a separate entity from the mind but works collaboratively to produce well-being.[47] Focusing on pain or catastrophic pain-related thoughts increase the perceived intensity of the sensation.[48 49] Indeed, it is interesting to note that in this study, participants used mind management to control the sensation of symptoms. Furthermore, positive and optimistic individuals are prone to recognise meaning in their lives and more willing to accept pain, subsequently returning more quickly to normal functioning levels.[33 50] Undoubtedly, a positive attitude was a central narrative in this study. In our study, maintaining a positive self-image using cognitive approaches such as positive self-statements and social downward comparison were reported. Consistent with previous studies, we suggest that when individuals face threatening situations such as chronic pain or long-term disease, social comparison strategies are adopted as a means of coping and adaptation.[51–54] Frieswijk et al reported that individuals may experience a less negative effect living with persistent symptoms after comparison with others in worse situations (downward comparison), evoking feelings of superiority and empowerment.[55]

To date, there are limited studies on religious coping in populations affected by rheumatic disorders.[56] Our findings indicate that religion and spirituality is an important form of coping for the faithful in our study sample, consistent with studies reporting a tendency to seek spiritual guidance from a higher power during time of illness and to pray for health-related concerns.[57–60] It is hypothesised that religion may enhance a sense of control over

disturbing events.[61] For example, for the faithful, praying may be a means of exerting some influence over the course of chronic adversity by asking for divine intervention or to gain strength to bear hardship.[61] Therefore, in the context of long-term chikungunya disease, a chronic illness with no curative treatment, religious coping may enable individuals to gain a sense of personal mastery over the disease and accept its chronic nature.

The findings of this qualitative study demonstrate that adopting adaptive coping strategies directed at retaining a sense of control of oneself and one's life is significant in living a valued life despite the continuing presence of rheumatic symptoms after CHIKV infection. Conceptually, having a sense of control constitutes the opposite end of powerlessness and thus more likely to lessen psychological distress.[62] Findings highlight the need for healthcare professionals in supporting individuals to recognise their strengths and retaining or regain a sense of control with interventions directed at enhancing adaptive coping strategies and resilience, alongside supportive care aimed at reducing rheumatic symptoms. Interventions such as cognitive–behavioural therapies and mindfulness training exercises for example, acceptance therapy, cognitive behavioural stress management and integrative body-mind management may provide individuals with the necessary tools to adopt an attitude of openness and acceptance towards one's experience.[63 64] Rather than continuously struggling with the pain, discomfort and thoughts related to persistent symptoms. Hence, strengthening or regain an individual's sense of competence and control by fostering adaptive coping skills and resilience. At the same time, this relates to positive health outcomes in terms of QoL.[64] By comparing intervention and matched control groups, future research could determine how these interventions affect coping strategies and health outcomes in clinical settings.

Our study has several strengths and limitations. This study is the first of its kind. To date, only one study explored coping strategies employed by individuals affected by chikungunya.[18] However, this study focused mainly on medical coping strategies including medicine use and uptake of medical care. Whereas in contrast our study focused on adaptive coping strategies that are effective at benefiting or positively affect the lives of individuals, taking into consideration that there are currently no effective medicines and treatment options for long-term chikungunya disease. To ensure rigour, a detailed description of the context, methods and results was provided, facilitating judgement of validity and transferability.[65 66] The explorative nature of the study supported in the understanding of the coping strategies more broadly and in context. This study is limited by the fact that all participants were Afro-Caribbean, and therefore, our findings may not be inclusive of the views of those of other ethnicities and cultures. In addition, the majority of our study sample may be categorised as having moderate to high socioeconomic status,[67] this profile has been reported to foster adaptive coping strategies,[68 69] including higher

levels of pain acceptance and resilience.[70 71] We assume that this was no major issue, since the aim of this study was to gain in-depth insights of the personal adaptive coping strategies employed by individuals affected by long-term chikungunya disease. Furthermore, coping is a dynamic process, which may change during the course of disease and has to be addressed in future longitudinal studies. In addition, participants were not medically assessed and the majority were women. However, the reported symptoms are in agreement with the current knowledge of long-term chikungunya disease sequelae,[5 8 9] in which woman are over-represented.[72] Despite these limitations, the findings of this study contribute to the limited body of literature about living with long-term chikungunya disease for which there are literally almost no data on coping strategies.

In closing, as there is no curative treatment for long-term chikungunya disease, affected individuals should be assisted to cope and overcome the disease in an effective way. More qualitative research is needed to explore the adaptive coping strategies employed and how these can be maximised to support well-being, especially among lower socioeconomic populations, who may face even more challenges in living and coping with long-term chikungunya disease.

**Acknowledgements** The authors would like to thank all participants for their willingness and active participation in our research, in the COVID-19 pandemic.

**Contributors** CD and AB conceived the study. CD was responsible for overseeing the project and carried out the interviews under supervision of AB. CD and AB were involved in data analysis. CD wrote the first draft of the manuscript. AB, AJD, AT and IG helped with the revision of the manuscript. All authors have agreed on the final version. AB accepts full responsibility for the work and/or the conduct of the study, had access to the data, and controlled the decision to publish.

**Funding** This work was supported by the Dutch Research Council (NWO) grant number ALWCA.2016.021.

**Competing interests** None declared.

**Patient and public involvement** Patients and/or the public were involved in the design, or conduct, or reporting, or dissemination plans of this research. Refer to the Methods section for further details.

**Patient consent for publication** Not applicable.

**Ethics approval** This study involves human participants and was approved by the Medical Ethical Committee of the Saint Elisabeth Hospital in Curaçao (Reference number: 2015-00). Participants gave informed consent to participate in the study before taking part.

**Provenance and peer review** Not commissioned; externally peer reviewed.

**Data availability statement** Data are available on reasonable request.

ORCID iDs
Churnalisa Doran http://orcid.org/0000-0002-9875-2758
Adriana Tami http://orcid.org/0000-0002-1918-9144

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
