## [Reviewer comments · BMJ Open]

ARTICLE DETAILS

TITLE (PROVISIONAL)	Adaptive coping strategies among individuals living with long-term chikungunya disease: A qualitative study in Curaçao
AUTHORS	Doran, Churnalisa; Duits, AJ; Gerstenbluth, Izzy; Tami, Adriana; Bailey, Ajay

VERSION 1 – REVIEW

REVIEWER	Natalia Romero Universidad Internacional del Ecuador, Quito, School of Medicine
REVIEW RETURNED	21-Aug-2023

GENERAL COMMENTS	This is a very interesting paper that explores which adaptive coping strategies are employed to manage persistent rheumatic symptoms in daily life, and shows relevance, and reliability. I'd like to highlight general and specific comments. General comment It is not clear the content of this paper and this other one: "Health-related impact on quality of life and coping strategies for chikungunya: A qualitative study in Curaçao" https://doi.org/10.1371/journal.pntd.0005987. I would like to recommend the authors to point out the main contribution of the previous publication and the contribution of this new article. Specific comments Abstract section Conclusions, in general, are not according to results. Some outcome statements could be conclusions. ("...the participants expressed a common desire for having a sense of being in control, to not let the disease take over, focusing on the positive in their lives, and finding strength and remain hopeful") In conclusion the words "To live a normal life" seem confusing. Normal life can mean different things to different people, as it is highly subjective and can be influenced by cultural, societal, or personal factors, and the main objective of this paper was not to characterise a "normal life". Lines 46-47: This statement seems confusing or incomplete: "Subsequently, improve health-related quality of life when rheumatic symptoms persist following chikungunya virus infection." Methods section Design The document does not include the theoretical framework on which the researchers define the theoretical categories. Only in Results section appears four themes, but they do not have a consistent theoretical basis. Please clarify. Moreover, without this prior positioning of knowledge, it is not possible to consider whether or not there were emergent categories. Participants and recruitment Line 118-119: The sample seems to be defined based on four
--

	characteristics (persistent rheumatic symptoms, pain intensity, gender, and age). If it is correct, please clarify these doubts:  1. Were these characteristics considered as participant profiles for the analysis? The methodology and results do not show this. If the authors did not consider the four characteristics as criteria for the narrative analysis it should be clear in the Methodology (participant sections, analysis). 2. The previous point is related with my next questions: How was reached the saturation? What were the purposive criteria to select the participants? 3. What were the exclusion criteria? 4. How were contacted the participants? Who contacted with them? 5. On the other hand, Table 1 presents seven characteristics of the participants, but anyone of them has been considered as criteria for the analysis. Again, it is confusing, because it's not clear if the profile of the participants was considered to structure the narrative analysis. Line 138: Which were the criteria to evaluate the adequacy of the sample size? The paper refers only to the number of in-depth interviews. Results section This section should be revised based on the comments made in Methodology. Line 216: The authors have used the words "normal life".  1. This issue has not been defined, nor has it been included in the theoretical categories. 2. This topic is considered as subjective, and it is no clear its contribution and use in the objective of this study. Discussion section This section also should be revised. I would like to recommend review these papers: https://doi.org/10.1515/sjpain-2021-0190 https://doi.org/10.1002/ejp.1062 https://doi.org/10.1007/978-3-319-22858-7_8 https://doi.org/10.1007/s10067-019-04919-1
--	--

VERSION 1 – AUTHOR RESPONSE

Reviewer 1

Comments:

1. This is a very interesting paper that explores which adaptive coping strategies are employed to manage persistent rheumatic symptoms in daily life, and shows relevance, and reliability.

Author's response: *We thank the reviewer for the appreciation.*

General comment

1. It is not clear the content of this paper and this other one: "Health-related impact on quality of life and coping strategies for chikungunya: A qualitative study in Curaçao"
<https://doi.org/10.1371/journal.pntd.0005987>. I would like to recommend the authors to point out the main contribution of the previous publication and the contribution of this new article.

Author's response: *We thank the reviewer for raising this point. Accordingly, the following text has been included in the Discussion section:*

-Lines 363-369: "To date only one study explored coping strategies employed by individuals affected by chikungunya. However, this study focused mainly on coping strategies including medicine use and uptake of medical care. Whereas in contrast our study focused on adaptive coping strategies that are effective at benefitting or positively affect the lives of individuals, taking into consideration that there are currently no effective medicines and treatment options for long-term chikungunya disease."

Specific comments

2. Abstract section

Conclusions, in general, are not according to results. Some outcome statements could be conclusions. ("...the participants expressed a common desire for having a sense of being in control, to not let the disease take over, focusing on the positive in their lives, and finding strength and remain hopeful").

Author's response: *We thank the reviewer for raising this point. Accordingly, the sentence "...the participants expressed a common desire for having a sense of being in control, to not let the disease take over, focusing on the positive in their lives, and finding strength and remain hopeful", has been removed and placed in the "Conclusion" section of the "Abstract".*

3. In conclusion the words "To live a normal life" seem confusing. Normal life can mean different things to different people, as it is highly subjective and can be influenced by cultural, societal, or personal factors, and the main objective of this paper was not to characterise a "normal life".

Author's response: *We thank the reviewer for raising this point. The phrase "To live a normal life" has been replaced by "To live with long-term chikungunya disease".*

4. Lines 46-47: This statement seems confusing or incomplete: "Subsequently, improve health-related quality of life when rheumatic symptoms persist following chikungunya virus infection."

Author's response: *We thank the reviewer for raising this point. Accordingly, the statement has been adapted:*

-Lines 45-47: "Subsequently, these interventions may improve health-related quality of life when rheumatic symptoms persist following chikungunya virus infection".

Methods section

5. Design

The document does not include the theoretical framework on which the researchers define the theoretical categories. Only in Results section appears four themes, but they do not have a consistent theoretical basis. Please clarify. Moreover, without this prior positioning of knowledge, it is not possible to consider whether or not there were emergent categories.

Author's response: We thank the reviewer for raising this point. We have analysed the data using thematic analysis inductively to provide a rich and detailed, yet complex account of the data. Thematic analysis does not require a pre-existing theoretical framework, unlike any other methods for qualitative data analysis. We have included the following text in the "Data analyses" section:

-Lines 150-151: "The data were analyzed using thematic analysis. Thematic analysis is data-driven and therefore does not require a pre-existing theoretical framework, unlike any other methods for qualitative data analysis."

-Lines 158-159: "The analysis process followed an inductive and iterative approach, which is a process of coding the data without trying to fit it into a pre-existing theoretical frame."

Participants and recruitment

Line 118-119: The sample seems to be defined based on four characteristics (persistent rheumatic symptoms, pain intensity, gender, and age). If it is correct, please clarify these doubts:

6. Were these characteristics considered as participant profiles for the analysis? The methodology and results do not show this. If the authors did not consider the four characteristics as criteria for the narrative analysis it should be clear in the Methodology (participant sections, analysis).

Author's response: We thank the reviewer for raising this point. We did not consider the four characteristics as criteria for the narrative analysis and have included the following text in the "Data analyses" section to remove any confusion of the reader:

-Lines 148-149: "The profile of the participants including the purposive sampling criteria and demographic data was not considered to structure or analyse the narratives."

7. The previous point is related with my next questions: How was reached the saturation? What were the purposive criteria to select the participants? What were the exclusion criteria?

Author's response: We thank the reviewer for raising these points.

To make clear how saturation was reached, the following text has been included in the "Data collection" section:

-Lines 141-143: "Saturation was considered to have been met when probing gained no additional information or insights, signifying that further data collection is redundant and an adequate sample size is reached."

To make clear what the purposive criteria were to select the participants, the following text has been adapted in the "Participants and Recruitment" section:

-Lines 112-117: "Maximum variation purposive sampling using cohort records from a follow-up study conducted between July 2019 and March 2020 was employed to identify potential participants based on persistent rheumatic symptoms (arthralgia, swelling, stiffness, cramps and/or locking in one or more joints and/or myalgia), pain intensity score (0-10) of at least 1, gender and different age, to allow for a wide range of experiences and adaptive coping strategies employed."

To make clear what the exclusion criteria was, the following text has been adapted in the “Participants and Recruitment” section:

-Lines 120-124: “The eligible patients were contacted by CD (first author) and invited to participate by telephone on the basis they met the inclusion criteria of still experiencing persistent rheumatic symptoms following CHIKV infection, at the time of contact, if not they were excluded. Out of the 29 individuals (19 women and 10 men) approached, 6 men declared being recovered from rheumatic symptoms and were excluded.”

8. How were contacted the participants? Who contacted with them?

Author’s response: *We thank the reviewer for raising this point. Accordingly, the following text has been included in the “Participants and Recruitment” section:*

-Line 120: “The eligible patients were contacted by CD (first author) and invited to participate by telephone...”

9. On the other hand, Table 1 presents seven characteristics of the participants, but anyone of them has been considered as criteria for the analysis. Again, it is confusing, because it’s not clear if the profile of the participants was considered to structure the narrative analysis.

Author’s response: *We thank the reviewer for raising this point. The profiles of the participants were not used to structure or analyse the data. Accordingly, the following text have been adapted or included in the “Data analyses” section and Discussion:*

-Lines 148-149: “The profile of the participants including the purposive sampling criteria and demographic data was not considered to structure and analyse the narratives.”

-Lines 374-379: “In addition, the majority of our study sample may be categorized as having moderate to high socio-economic status, this profile has been reported to foster adaptive coping strategies, including higher levels of pain acceptance and resilience. We assume that this was no major issue, since the aim of this study was to gain in-depth insights of the personal adaptive coping strategies employed by individuals affected by long-term chikungunya disease.”

10. Line 138: Which were the criteria to evaluate the adequacy of the sample size? The paper refers only to the number of in-depth interviews.

Author’s response: *We thank the reviewer for raising this point. The following text has been included in the “Data collection” section:*

-Lines 141-143: “Saturation was considered to have been met when probing gained no additional information or insights, signifying that further data collection is redundant and an adequate sample size is reached.”

Results section

12. This section should be revised based on the comments made in Methodology.

Author response: We thank the reviewer for this recommendation. However, by using a thematic approach to analyse the data and the coping experiences of the participants is the main focus of our study, rather than a theoretical framework or participants profile, we have not revised the Results.

13. Line 216: The authors have used the words “normal life”. 1. This issue has not been defined, nor has it been included in the theoretical categories. 2. This topic is considered as subjective, and it is not clear its contribution and use in the objective of this study.

Author’s response: We thank the reviewer for raising these points. The words “normal life” has been removed throughout the manuscript.

Discussion section

14. This section also should be revised. I would like to recommend review these papers:

<https://doi.org/10.1515/sjpain-2021-0190>

<https://doi.org/10.1002/ejp.1062>

https://doi.org/10.1007/978-3-319-22858-7_8

<https://doi.org/10.1007/s10067-019-04919-1>

Author’s response: We thank the reviewer for this recommendation. However, we have not revised the results (see author’s response to reviewers point 12). Therefore, it was not necessary to revise the Discussion section. The recommended papers have been reviewed and the papers regarding resilience have been included as additional references in the Discussion section.

VERSION 2 – REVIEW

REVIEWER	Natalia Romero Universidad Internacional del Ecuador, Quito, School of Medicine
REVIEW RETURNED	15-Jan-2024
GENERAL COMMENTS	Thanks to the authors for the responses to comments and the work on the new version. This is an interesting study that may help people with chikungunya consequences.